# Trends in breast cancer incidence in Ho Chi Minh City 1996–2015: A registry-based study

**Dung X. Pham[1,2◍], Thao-Quyen H. Ho[3‡], Tung D. Bui[4‡], Lan T. Ho-Pham 🅾[5,6◍]*, Tuan V. Nguyen[6,7,8,9◍]**

**1** Ho Chi Minh City Oncology Hospital, Ho Chi Minh City, Vietnam, **2** Department of Oncology, Pham Ngoc Thach University of Medicine, Ho Chi Minh City, Vietnam, **3** Department of Training and Scientific Research, University Medical Center, Ho Chi Minh City, Vietnam, **4** Department of Healthcare Directions, Ho Chi Minh City Oncology Hospital, Ho Chi Minh City, Vietnam, **5** BioMedical Research Center, Pham Ngoc Thach University of Medicine, Ho Chi Minh City, Vietnam, **6** Bone and Muscle Research Group, Ton Duc Thang University, Ho Chi Minh City, Vietnam, **7** Bone Biology Division, Garvan Institute of Medical Research, Sydney, New South Wales, Australia, **8** St Vincent's Clinical School, UNSW Medicine, UNSW Sydney, Sydney, New South Wales, Australia, **9** School of Biomedical Engineering, University of Technology Sydney, Sydney, New South Wales, Australia

◍ These authors contributed equally to this work.
‡ These authors also contributed equally to this work.
* thuclanhopham@pnt.edu.vn

## Abstract

The burden of breast cancer in Vietnam has not been documented. This study sought to estimate the incidence of breast cancer in Ho Chi Minh City, the largest economic center of Vietnam, from 1996 to 2015. This was a population-based study using the Ho Chi Minh City Cancer Registry as a source of data (coverage period: 1996–2015). The Registry adopted the International Classification of Diseases for Oncology, 3rd Edition for the classification of primary sites and morphology, and guidelines from the International Agency for Research on Cancer and the International Association of Cancer Registries. Using the population statistics from census data of Ho Chi Minh City, the point incidence of breast cancer for 5-year period was estimated. Based on the national population, we calculated the age-standardized rate (ASR) of breast cancer between 1996 and 2015. Overall 14,222 new cases of breast cancer (13,948 women, or 98%) had been registered during the 1996–2015 period; among whom, just over half (52%) were in the 2nd stage and 26% in the 3rd and 4th stages. In women, the median age at diagnosis was 50 years and there was a slight increase over time. The ASR of breast cancer during the 2011–2015 period was 107.4 cases per 100,000 women, representing an increase of 70% compared to the rate during the 1996–2000 period. In men, there was also a significant increase in the ASR: from 1.13 during the 1996–2001 period to 2.32 per 100,000 men during the 2011–2015 period. These very first data from Vietnam suggest that although the incidence of breast cancer in Vietnam remains relatively low, it has increased over time.

**Data Availability Statement:** Data from all registries was collected and assessed based on guidelines from the International Agency for Research on Cancer and the International Association of Cancer Registries, adapted to a low

and middle-income context. Registry data are stored in a computerized data at the Oncology Hospital of Ho Chi Minh City. The dataset includes the following variables: age, sex, diagnosis, year of diagnosis, cancer stage, treatment, and survival status. All authors had no special access privileges in accessing these datasets which other interested researchers would not have. Age-and-gender population statistics were obtained from census data managed by the General Statistics Office (GSO) of Ho Chi Minh City. Age-and-gender population statistics in 1999 for Vietnam was obtained from the Bureau of Statistics of Vietnam. There is no URL for the population data. However, request for data access can be made to the General Statistics Office of Ho Chi Minh City, contact through email tdnlinh@ump.edu.vn.

**Funding:** The authors received no specific funding for this work.

**Competing interests:** The authors have declared that no competing interests exist.

## Introduction

Breast cancer is the most common cancer in women worldwide. In the United States alone, projected statistics in 2019 showed that approximately 30% of all cancers in women were attributable to breast cancer [1, 2]. Moreover, in the Asia Pacific region, incomplete data indicate that breast cancer was the most common type of cancer, accounting for 18% of total cancers in women [3]. In absolute number, the International Agency for Research on Cancer (IARC) estimated that in 2018 alone, 2.1 million women were diagnosed with breast cancer, and 627 women died from the disease [4]. With the rapid aging of the population worldwide, the burden of breast cancer is expected to increase in the future.

There is a geographic disparity in the distribution of breast cancers. At present, the incidence of breast cancer in Asian populations is lower than in white populations. The age-adjusted incidence rate of cancer in Asian populations was 29 per 100,000 women, which is about a third of that in the American population (~93 per 100,000 women); however, the risk of mortality from breast cancer in Asians is higher than that in women of European descent [5]. More interestingly, Asian women tend to have breast cancer at a younger age than their white counterparts: 47% of women with a diagnosis of breast cancers aged 50 years or younger, but this proportion was 33% in the world [3].

Although there have been extensive studies on breast cancer in economically advantaged countries, the incidence, prevalence, and risk factors for breast cancer in Vietnam have not been well documented. Results from one case-control study found that breast density, age at first menarche, menopause status, number of pregnancies, number of babies born, hormone use and no physical activities were significantly associated with breast cancer in Vietnamese women [6]. Vietnam is the 15th most populous country in the world, with a population of 97 million [7]. Almost 23% of the population aged 50 years and older. However, until now, there has been no systematic documentation of the incidence of breast cancer in Vietnam over the past 20 years. In this study, we sought to estimate the incidence of breast cancer in Ho Chi Minh City, the largest city in Vietnam. Our result provides important data concerning the burden of breast cancer in the country that is rapidly transiting from an agricultural economy to a modern economy.

## Study design and methods

The anonymized data for this study were extracted from the Ho Chi Minh City Cancer Registry. The Registry was established in 1990 to document all diagnosed cancer cases in the City. Cancer patients admitted to any hospital in the City were ascertained and checked for possible duplication. The coverage period was from January 1, 1996 to December 31, 2015. This study was restricted to people who were identified as residents of Ho Chi Minh City on their patient records. We focused on Ho Chi Minh City, because (i) it is the largest center of commerce in the country, with a population of 8.2 million (2014 statistics); (ii) the ascertainment and documentation of cancers in the City is more complete than any other provinces in the country; and (iii) the City offers an opportunistic setting for studying the burden of cancers in a transitional population. The study was approved by the Ethics Committee of the Oncology Hospital of Ho Chi Minh City. Because all data were anonymized no individual patient consent was required.

The Ho Chi Minh City Cancer Registry adopted the International Classification of Diseases for Oncology, 3rd Edition (ICDO-3) for the classification of primary sites and morphology, and guidelines from the International Agency for Research on Cancer and the International Association of Cancer Registries. Based on the ICDO-3, we identified breast cancer cases from

1 January 1996 to 31 December 2015 inclusive. The identification was further ascertained by tumor site code, morphology code, and behavior type.

Data from all registries was collected and assessed based on guidelines from the International Agency for Research on Cancer and the International Association of Cancer Registries, adapted to a low and middle-income context. Data was validated through clinical records, coded, and verified according to guidelines.

Age-and-gender population statistics were obtained from census data managed by the General Statistics Office (GSO) of Ho Chi Minh City. Population statistics were available for 1999, 2004, 2009, and 2014. Age-and-gender population statistics in 1999 for Vietnam were obtained from the Bureau of Statistics of Vietnam.

Using the population statistics of Ho Chi Minh City, we computed the point incidence rate of breast cancer (per 100,000 population) for each 5-year interval: 1996–2000, 2001–2005, 2006–2010, and 2011–2015 inclusive. The reason for aggregating 5-year data was to improve the stability of statistical estimates. We used the direct method of standardization to calculate the age-standardized rate (ASR), by applying the age-specific rates observed in a period to the national population in 1999. In this approach, the ASR can be thought of as a weighted average rate, with the weights being the proportion of the national population in each age group.

We employed a logistic joint point regression model [8] to identify temporal changes in the incidence of breast cancer over the coverage period. Assuming that the annual incidence of cases in the population follows the Binomial distribution, the logistic joint point regression uses the logit of the incidence rate as the dependent variable to identify the best fit for joint points (i.e., inflexion points) at which there is a significant change in trends. Based on exploratory analysis, we allowed maximum of 2 points in the model. The analyses were conducted using the R Statistical Environment [9] and "ljr" package [8].

## Results

Between January 1, 1996 and December 31, 2015, 14,222 new cases of breast cancer (13,948 women, or 98%) had been registered in the Registry (Table 1). Almost 100% of cancers were classified as malignant. Based on the data on stage (n = 422 women), just over half (52%) were in the 2nd stage, and 26% of cases were in the 3rd and 4th stages.

The median age at diagnosis was 50 years (IQR: 43–59) and 55 years (IQR: 47–65) for women and men, respectively. In women, there was a slight but statistically significant increase in the average age of diagnosis of breast cancer between 1996 and 2015. Between 1996 and 2000, the median age at diagnosis was 49, and this was increased to 51 yrs during 2011 and 2015 (P = 0.006). Between 1996 and 2000, 48.8% of women with a breast cancer diagnosis aged 50 years and older, and this proportion was increased to 55.8% during 2011 and 2015 (Table 2).

Joint point regression analysis identified 2 breakpoints, 1999 and 2005, in the incidence of breast cancer in women throughout 1996–2015 (Fig 1). The average rates of increase in the incidence during the period of 1996–1999, 2000–2005, and 2006–2015 were 0.019, 0.001, and 0.013, respectively.

In women, the 5-year age-standardized incidence rate of breast cancer was 62.2 cases per 100,000 population during the 1996–2000 period, and this was progressively increased to 107.4 during the 2011–2015 period, representing a 70% increase over the 20-year period. In men, there was also a significant increase in the age-standardized incidence rate: from 1.13 during the 1996–2001 period to 2.32 during the 2011–2015 period, representing an increase of 2.1-fold (Table 3).

**Table 1. Clinical characteristics of 13948 women and 274 men with breast cancer in Ho Chi Minh City, 1996–2015.**

| | Women (n = 13948) | Men (n = 274) | Total (n = 14222) |
|---|---|---|---|
| **Type of cancer** | | | |
| In situ | 44 (0.3%) | 0 (0%) | 44 (0.3%) |
| Malignant | 13892 (99.6%) | 274 (100%) | 14166 (99.6%) |
| Uncertain | 12 (0.1%) | 0 (0%) | 12 (0.1%) |
| **Stage of cancer** | | | |
| I | 87 (0.6%) | 0 (0%) | 87 (0.6%) |
| II | 223 (1.6%) | 4 (1.5%) | 227 (1.6%) |
| III | 69 (0.5%) | 0 (0%) | 69 (0.5%) |
| IV | 43 (0.3%) | 1 (0.4%) | 44 (0.3%) |
| Unknown | 13525 (97.0%) | 269 (98.2%) | 13794 (97.0%) |
| **Base of Diagnostic** | | | |
| Biochemical/Immuno tests | 6 (0.0%) | 0 (0%) | 6 (0.0%) |
| Clinical examination only | 40 (0.3%) | 1 (0.4%) | 41 (0.3%) |
| Clinical test | 1066 (7.6%) | 25 (9.1%) | 1091 (7.7%) |
| Cytology/Haematology | 1284 (9.2%) | 19 (6.9%) | 1303 (9.2%) |
| Exploratory surgery | 270 (1.9%) | 6 (2.2%) | 276 (1.9%) |
| Histology of Metastasis | 214 (1.5%) | 6 (2.2%) | 220 (1.5%) |
| Histology of Primary | 11033 (79.1%) | 217 (79.2%) | 11250 (79.1%) |
| Missing | 35 (0.3%) | 0 (0%) | 35 (0.2%) |
| **Treatment** | | | |
| Chemotherapy | 1523 (10.9%) | 27 (9.9%) | 1550 (10.9%) |
| Hormonotherapy | 38 (0.3%) | 1 (0.4%) | 39 (0.3%) |
| No treatment | 2527 (18.1%) | 56 (20.4%) | 2583 (18.2%) |
| Others | 80 (0.6%) | 3 (1.1%) | 83 (0.6%) |
| Radiotherapy | 96 (0.7%) | 24 (8.8%) | 120 (0.8%) |
| Surgery | 9543 (68.4%) | 161 (58.8%) | 9704 (68.2%) |
| Missing | 141 (1.0%) | 2 (0.7%) | 143 (1.0%) |
| **Status** | | | |
| Alive | 13437 (96.3%) | 260 (94.9%) | 13697 (96.3%) |
| Dead | 101 (0.7%) | 4 (1.5%) | 105 (0.7%) |
| Unknown | 354 (2.5%) | 10 (3.6%) | 364 (2.6%) |
| Missing | 56 (0.4%) | 0 (0%) | 56 (0.4%) |

Note: number in bracket represents the column-wise percentage.

## Discussion

It has been projected that economically less developed countries are going to bear a greater burden of breast cancer than more developed countries [10]. However, research on risk factors for and incidence of breast cancer in population among less developed countries have been scarce. In this study, by using a well-characterized registry-based data of the largest city in Vietnam, we have shown that over the past 20 years, the incidence of breast cancer had increased by 70% and that the increase was mainly attributable to those age groups of 50 and 70. These very first results from Vietnam deserves further elaboration.

It seems clear that the age-standardized incidence of breast cancer in this study is lower than that in white populations. For instance, in the United States, between 2009 and 2016, the age-standardized incidence rate was approximately 200 per 100,000 woman-years [11] which is higher than in Australia (~131 per 100,000 women) [12]. In China, the age-standardized

**Table 2. Age distribution of breast cancer by age group and by sex in Ho Chi Minh City, Vietnam (1996–2015).**

| Gender | Age group | 1996–2000 | 2001–2005 | 2006–2010 | 2011–2015 |
|---|---|---|---|---|---|
| **Women** | <30 | 24 (1.3) | 33 (1.2) | 45 (1.2) | 81 (1.5) |
| | 30–39 | 256 (13.4) | 368 (12.9) | 457 (12.0) | 622 (11.6) |
| | 40–49 | 702 (36.7) | 1090 (38.3) | 1310 (34.3) | 1668 (31.1) |
| | 50–59 | 431 (22.5) | 726 (25.5) | 1194 (31.3) | 1749 (32.6) |
| | 60–69 | 302 (15.8) | 401 (14.1) | 519 (13.6) | 834 (15.5) |
| | 70–79 | 153 (8.0) | 182 (6.4) | 224 (5.9) | 326 (6.1) |
| | 80+ | 47 (2.5) | 48 (1.7) | 70 (1.8) | 86 (1.6) |
| | All ages | 1915 | 2848 | 3819 | 5366 |
| **Men** | <30 | 0 | 0 | 2 (2.4) | 1 (1.0) |
| | 30–39 | 3 (9.7) | 10 (16.9) | 4 (4.7) | 7 (7.1) |
| | 40–49 | 10 (32.3) | 9 (15.3) | 22 (25.9) | 24 (24.2) |
| | 50–59 | 8 (25.8) | 12 (20.3) | 26 (30.6) | 29 (29.3) |
| | 60–69 | 7 (22.6) | 13 (22.0) | 10 (11.8) | 23 (23.2) |
| | 70–79 | 2 (6.5) | 10 (16.9) | 18 (21.2) | 14 (14.1) |
| | 80+ | 1 (3.2) | 5 (8.5) | 3 (3.5) | 1 (1.0) |
| | All ages | 31 | 59 | 85 | 99 |

Note: numbers in brackets represent the sex-specific percent of the total for each 5-year interval.

incidence of breast cancer was observed at 28.4 per 100,000 women [13]. In Vietnam's neighboring country Thailand, the incidence rate was estimated to be 31.2 per 100,000 woman-years [14]. In our study, the annualized age-standardized incidence rate was 21.5 per 100,000 women throughout 2011 and 2015, which can still be considered low relative to populations in more economically developed countries.

We found that the incidence of breast cancer in our cohort had been increasing over time, and this trend is consistent with previous observations [15–17]. In the 1996–2000 period, the age-standardized incidence rate was 12.4 per 100,000 women which is comparable with a previous estimate [18]. However, 20 years later, this incidence was increased by almost two-fold. It has been projected that the incidence of all cancers in the two major cities of Vietnam will be increased by approximately 17% over the next 5 years [19]. The increase in breast cancer incidence rates has been found in China, Japan, and Thailand [20]. This increase in the incidence plus the large population sizes in Asia imply that more cancer cases will be observed in Asian populations than in white populations. The increase in life expectancy and the aging of the population contribute three-fifths and the increased age-standardized rates contribute two-fifths of future trend [21]. If this assumption holds and given the improvement in life expectancy in Vietnamese women (~79.4 years) [22], it is expected that breast cancer will impose a heavy burden in Vietnam shortly. The increase in cancer incidence could also reflect better registration of cases and/or public awareness over the coverage period.

The age structure of cancer cases in our cohort merits a comment. We found that the median age at diagnosis of breast cancer was 50 years (in women), which is almost identical to the median age at diagnosis of breast cancer among Singaporean [23] and Korean [24] women. In economically more developed countries, the average age at breast cancer diagnosis was ~54 years [25] or 59 years [26]. Thus, our finding reaffirms the common 'law' that Asian women tended to have breast cancer at a younger age than their white counterparts.

Almost 100% of breast cancer tumors in this study were in the invasive stage, only 0.3% was in situ, and 26% of cases were diagnosed at late stage III and IV. The proportion of in-situ

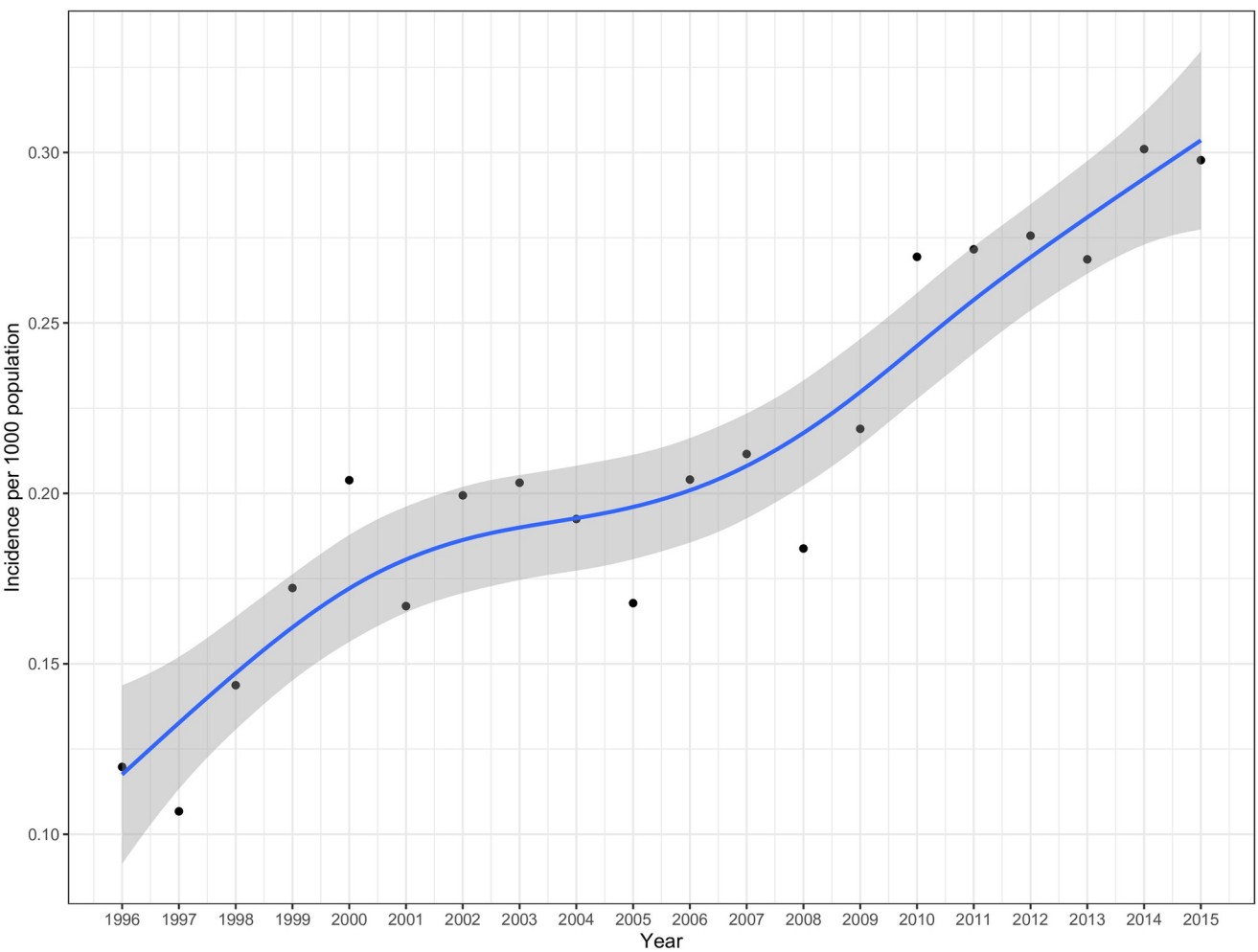

**Fig 1. Estimated annual incidence rate of breast cancer (per 1000 population) in women during the 1996–2015 period in Ho Chi Minh City.**

tumors was 16% in the United States [27] and 14% in South Korea [24]. Delayed diagnosis of cancer is a major factor that contributes to the increased risk of premature death, lower cancer survival and increases the burden of cancer. Indeed, the overall 5-year relative survival rate is 99% for localized disease, 85% for regional disease and 27% for distant-stage disease [28]. Our finding implies that a more aggressive screening strategy for identifying cancer cases earlier is warranted.

**Table 3. Unadjusted and standardized incidence rate (per 100,000 persons over 5 years) of breast cancer in Ho Chi Minh City (1996–2015) stratified by gender.**

| Gender | Estimate | 1996–2000 | 2001–2005 | 2006–2010 | 2011–2015 |
|---|---|---|---|---|---|
| Women | Unadjusted | 73.3 (1.67) | 89.7 (1.68) | 102.5 (1.66) | 137.1 (1.87) |
| | Standardized | 62.2 (0.38) | 72.5 (0.41) | 79.3 (0.43) | 107.4 (0.51) |
| | Rate Ratio | 1.00 | 1.20 | 1.30 | 1.70 |
| Men | Unadjusted | 1.28 (0.23) | 2.01 (0.26) | 2.47 (0.27) | 2.74 (0.28) |
| | Standardized | 1.13 (0.05) | 1.77 (0.06) | 2.01 (0.07) | 2.32 (0.08) |
| | Rate Ratio | 1.00 | 1.60 | 1.80 | 2.10 |

Note: numbers in brackets represent standard error.

Our findings should be interpreted within the context of the study's strengths and weaknesses. The data were ascertained from a well-developed registry that could capture total cancer incidence in Ho Chi Minh City. It is worth noting that virtually all breast cancer cases are treated at hospitals within the City, with a very small proportion being treated overseas, and the data were therefore likely complete. However, a caveat of the study is that we could not follow individual patients to ascertain their survival status, and as a result, we could not analyze the rate of mortality among these patients. Moreover, our finding concerning incidence may not be generalized to rural or non-urban areas where the incidence is expected to be lower than that in urban areas.

In conclusion, our registry-based data suggest that although breast cancer incidence in Ho Chi Minh City remained relatively low compared to white populations, there was an increasing trend up to 70% over the past 20 years. Our data also confirm that Vietnamese women tend to have breast cancer at younger ages compared to white women. These findings imply that breast cancer screening should be targeted women of younger ages.

## Acknowledgments

We gratefully acknowledge the assistance of the Ho Chi Minh City Oncology Hospital, Vietnam.

## Author Contributions

**Conceptualization:** Lan T. Ho-Pham, Tuan V. Nguyen.

**Data curation:** Dung X. Pham, Lan T. Ho-Pham.

**Formal analysis:** Dung X. Pham, Thao-Quyen H. Ho, Tung D. Bui, Lan T. Ho-Pham, Tuan V. Nguyen.

**Methodology:** Dung X. Pham, Lan T. Ho-Pham, Tuan V. Nguyen.

**Writing – original draft:** Dung X. Pham, Thao-Quyen H. Ho, Tung D. Bui, Lan T. Ho-Pham, Tuan V. Nguyen.

**Writing – review & editing:** Dung X. Pham, Lan T. Ho-Pham, Tuan V. Nguyen.

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
