## [Decision Letter · Decision Letter 0]

2 Sep 2020

PONE-D-20-16460

Trends in breast cancer incidence in Ho Chi Minh City 1996 - 2015: a registry-based study

PLOS ONE

Dear Dr. Ho-Pham,

Thank you for submitting your manuscript to PLOS ONE. After careful consideration, we feel that it has merit but does not fully meet PLOS ONE’s publication criteria as it currently stands. Therefore, we invite you to submit a revised version of the manuscript that addresses the points raised during the review process.

We look forward to receiving your revised manuscript.

Kind regards,

Chunyan He, ScD

Academic Editor

PLOS ONE

Journal Requirements:

2. In the ethics statement in the manuscript and in the online submission form, please provide additional information about the patient records used in your retrospective study. Specifically, please ensure that you have discussed whether all data were fully anonymized before you accessed them and/or whether the IRB or ethics committee waived the requirement for informed consent. If patients provided informed written consent to have data from their medical records used in research, please include this information.

3. For studies involving humans categorized by race/ethnicity authors should update outmoded terms and potentially stigmatizing labels to more current, acceptable terminology. Specifically, “Caucasian” should be changed to “white” or “of [Western] European descent” (as appropriate).

4. To comply with PLOS ONE submission guidelines, in your Methods section, please provide additional information regarding your statistical analyses. For more information on PLOS ONE's expectations for statistical reporting, please see https://journals.plos.org/plosone/s/submission-guidelines.#loc-statistical-reporting.

5.In your Data Availability statement, you have not specified where the minimal data set underlying the results described in your manuscript can be found. PLOS defines a study's minimal data set as the underlying data used to reach the conclusions drawn in the manuscript and any additional data required to replicate the reported study findings in their entirety. All PLOS journals require that the minimal data set be made fully available. For more information about our data policy, please see http://journals.plos.org/plosone/s/data-availability.

Reviewers' comments:

Reviewer's Responses to Questions

**Comments to the Author**

1. Is the manuscript technically sound, and do the data support the conclusions?

Reviewer #1: No

Reviewer #2: Yes

Reviewer #3: No

2. Has the statistical analysis been performed appropriately and rigorously? 

Reviewer #1: No

Reviewer #2: Yes

Reviewer #3: No

3. Have the authors made all data underlying the findings in their manuscript fully available?

Reviewer #1: No

Reviewer #2: Yes

Reviewer #3: Yes

4. Is the manuscript presented in an intelligible fashion and written in standard English?

Reviewer #1: No

Reviewer #2: Yes

Reviewer #3: No

5. Review Comments to the Author

Reviewer #1: The research was only a descriptive analysis of the number of newly discovered breast cancer patients, not a cohort study as described in the method. Data was not analysed nor discussed in any depths other than age standardized and age specific. More should be done in depth of lag time, time of diagnosis, diagnosis reason and age of diagnosis from these supposedly complete data from that long period of time that can yield so much more than a simple analysis like this. Plus, the screening strategy over that long period of time should be explained in details to rule out any bias and confoundings in the finding of increasing/decreasing incidence rate.

Reviewer #2: The author need to include the approved decision from Ethics Committee.

The author need to explain more how they do the standardized estimation (result in table 3) especially total of those who come for screening for cancel was not showed.

Some text error recorded e.g (end of page 5, table 3 - unadjusted and some where else)

Reviewer #3: Comments for article “Trends in breast cancer incidence in Ho Chi Minh City 1996 - 2015: a registry-based study”

This article deals with the evolution of breast cancer incidence in Ho Chi Minh City of Vietnam country between 1996 to 2015. This is an important issue, since such analyses enable a measurement of the burden for the health system, to make choices for health policy.

Nevertheless, such analyses require three major things that also need to address in this article

1. How far are the data reliable? How can we be sure of the exhaustivity of the data, from 1996 to 2015 in specified country? The effects observed may be just an effect of a better registration of the cancer cases. A detailed description of methods of registration in that country should be provided, to discuss the evolution of incidence rates.

2. To discuss the observed trends, we need information about health facilities in underline country and health policy concerning breast cancer.

3. The application of relevant statistical methods are major concern to get the precise results.

Therefore, last two points are lacking and make this work difficult to appreciate, although the subject is of real importance and manuscript comprises of a large data set but needs major revision in terms of data analysis, technical writing, sentence structure improvement, and removal of grammatical errors.

Major highlights are pointed out as follow:

Introduction

Line 79-80: “Vietnam is the 15th most populous country in the world, with a population of

97 million (2020 statistics). Please Cite reference here.

Additionally, breast cancer risk factors should also be addressed in introduction.

Study design and method

This section needs to include relevant statistical methods used for data analysis. It is recommended that use median and IQR for nonparametric data (age diagnosis) and use kruskal wallis test to measure the difference among age groups of diagnosed cases in case of scale measurements and use chi-square where percentages are compared. Further, Join-point regression can be used for trend analysis.

Line 115-116 “we computed the point incidence of breast cancer for each 5-year period… 2010, and 2011 - 2015 inclusive”. Briefly, mention how that incidence rate was calculated?

Line 118-119: “we calculated the age-standardized incidence rate for each of the 4 periods…” How age-standardized incidence rate was calculated?

Line 119-121: “We also employed a segmented Poisson regression model to estimate the change in the incidence of breast cancer over time. All statistical analyses were conducted using the R Statistical Environment….”.

It is recommended that use join-point regression technique and report your results in form of estimated annual percentage change (EAPC) with 95% UI. Please mention R version and package name that used for analysis.

Results

Line 125: findings (13,498 women, or 95%) and same in abstract, are inconsistent with results calculated in Table 1. (Table 1 has 13,948 women, 98%). Additionally report p-value of test difference and chi-square value.

Line 133-138, “The average at diagnosis was 52 years (SD 11.6) and 56.3 years (13.4) for women and men, respectively. In women, there was a slight but statistically significant increase …….(Table 2)”.

Firstly, data is nonparametric so authors should report median with IQR for age diagnosis, rather than mean and SD. Author reported median age in abstract but results section included mean age? Please be consistent. Authors reported that “in women statistically significant increase in the average age of diagnosis of breast cancer during 1996 and 2015….”. This difference look non-significant; please report p-value with test statistic value. Also, include these results in Table. Table 2 can be revised using joint-point regression estimate (EAPC with 95%UI for each duration).

Line:143-149: “Segmented regression indicated that there were two trends in the incidence of breast cancer in women: the first period occurred between ….(Figure 1). Further analysis showed that there was a statistically significant increase in the age-specific incidence of breast cancer over the period of 1996 and 2015, and the increase ….(Figure 2). In women, the increase in the age-specific incidence rate was observed among those aged…..”.

Join-point regression is widely used trend analysis technique and segmented regression is a part of it. Therefore, it is suggested that use main name of the technique for convenience of the readers.

Figure 1 and 2 , are not readable. A better presentation is needed here. Author should draw the trends across ages, years and cohort by year and age group (e.g. within age Group, within year and within cohort). Through these 3 figures Table 2 results can be well representative.

Line 160: Table 3. Briefly explain how the standardization was performed (ASR)?

Conclusion

Line 53-54: “These very first data from Vietnam suggest that although the incidence of breast cancer in Vietnam remains relatively low, it has increased over time, and that the increase was mainly attributable to those age groups of 50 and 70”.

Finally, authors concluded that “Our data also confirm that Vietnamese women tend to have breast cancer at younger ages compared to Caucasian women”. Younger ages? Conclusion is not consistent with the findings. (Further, most of the GBD studies reported that women breast cancer is more prevalent in older ages worldwide). Make any changes to the abstract that align with those made in the text.

Minor comments

Line 34: Revise sentence structure (In line with the related literature, for example, see following literature)

1. (Nguyen TP, Luu HN, Nguyen MV, Tran MT, Tuong TT, Tran CT, Boffetta P. Attributable Causes of Cancer in Vietnam. JCO global oncology. 2020 Feb;6:195-204.

2. Nguyen SM, Deppen S, Nguyen GH, Pham DX, Bui TD, Tran TV. Projecting cancer incidence for 2025 in the 2 largest populated cities in Vietnam. Cancer Control. 2019 Jul 22;26(1):1073274819865274.

3. Pham T, Bui L, Kim G, Hoang D, Tran T, Hoang M. Cancers in Vietnam—burden and control efforts: a narrative scoping review. Cancer Control. 2019 Jul 17;26(1):1073274819863802.)

Line 51: age-standardized incidence rate, replace it with ASR as its already mentioned in the abstract.

Line 55: those age groups of 50 and 70 years…

Line 60: 2019 showed that approx……..

Line 61: improper sentence structure.. What is meant by was also the most..?

Line 69: is lower than in Caucasian populations…….. remove “in”

Line 71 & 72. Mention reference 5 once after the completion of related information.

Line 73: women tends to have breast cancer in …….

Line 125: write as 1st Jan. 1996……

Line 127-128: Rewrite these lines

Line 133: word age is missing, modify as … The average age at diagnosis was 52 years ….

Line 145: later instead of latter

158: repetitive words in one sentence

Line 189: revise the sentence

198: unable to understand the statement

Line 202: Asian women tends to ….

Line 224: cancer incidence rates in Vietnam's urban population remain …..

6. PLOS authors have the option to publish the peer review history of their article (what does this mean?). If published, this will include your full peer review and any attached files.

Reviewer #1: No

Reviewer #2: No

Reviewer #3: No

---

## [Author Response · Author response to Decision Letter 0]

3 Nov 2020

Reviewer 1

"The research was only a descriptive analysis of the number of newly discovered breast cancer patients, not a cohort study as described in the method. Data was not analysed nor discussed in any depths other than age standardized and age specific. More should be done in depth of lag time, time of diagnosis, diagnosis reason and age of diagnosis from these supposedly complete data from that long period of time that can yield so much more than a simple analysis like this. Plus, the screening strategy over that long period of time should be explained in details to rule out any bias and confoundings in the finding of increasing/decreasing incidence rate." 

Authors: The Reviewer is correct that this study was a descriptive analysis of breast cancer incidence using a registry data base. Because the data were derived from a registry, it was not possible to have a complete clinical data such as reason for diagnosis or lag time. The data were not from a screening program. 

Although the study is descriptive in nature, the data are very important. These data represent the first documentation of breast cancer incidence from Vietnam. Because of its descriptive nature, we mainly employed descriptive statistical methods that are also used by virtually all previous studies. 

Reviewer 2 

"The author need to include the approved decision from Ethics Committee."

Authors: The study was indeed approved by the hospital ethics committee. We have now included the approval. 

"The author need to explain more how they do the standardized estimation (result in table 3) especially total of those who come for screening for cancel was not showed."

Authors: We have expanded the Methods section to describe the computation of age-standardized rate. 

"Some text error recorded e.g (end of page 5, table 3 - unadjusted and somewhere else)"

Authors: Thank you. We have gone through the manuscript and corrected all mis-spelling words. 

Reviewer 3 

"This article deals with the evolution of breast cancer incidence in Ho Chi Minh City of Vietnam country between 1996 to 2015. This is an important issue, since such analyses enable a measurement of the burden for the health system, to make choices for health policy. Nevertheless, such analyses require three major things that also need to address in this article."

Authors: Thank you for your positive remarks on our work. Indeed, the incidence of cancer in Vietnam has not been well documented. We hope that this paper provides data pertaining to the most important cancer (i.e. breast cancer) in Saigon, a major city in Vietnam, and that the data will help future planning of cancer prevention. 

"1. How far are the data reliable? How can we be sure of the exhaustivity of the data, from 1996 to 2015 in specified country? The effects observed may be just an effect of a better registration of the cancer cases. A detailed description of methods of registration in that country should be provided, to discuss the evolution of incidence rates."

Authors: We cannot really comment on the degree of accuracy of the data. However, under the assumption that patients affected by cancer seek medical attention, we can confirm that the registry captures virtually all cancer cases in Ho Chi Minh City. However, as the reviewer mentions, the increase in the incidence could be due to better ascertainment and public awareness, and this is now mentioned in the Discussion. We have now extended the description of the registry in the Methods section.

"2. To discuss the observed trends, we need information about health facilities in underline country and health policy concerning breast cancer."

Authors: In Ho Chi Minh City, all major hospitals provide care to patients affected by cancer. 

"3. The application of relevant statistical methods are major concern to get the precise results.

Therefore, last two points are lacking and make this work difficult to appreciate, although the subject is of real importance and manuscript comprises of a large data set but needs major revision in terms of data analysis, technical writing, sentence structure improvement, and removal of grammatical errors."

Authors: This is a descriptive study, and we have employed descriptive statistical methods. We consider that more advanced statistical methods are not necessary for this type of study. We have now gone through the manuscript again, and corrected all errors. 

"Major highlights are pointed out as follow:

Introduction

Line 79-80: "Vietnam is the 15th most populous country in the world, with a population of 97 million (2020 statistics). Please Cite reference here."

Authors: We have inserted a reference. 

"Additionally, breast cancer risk factors should also be addressed in introduction."

Authors: We consider that risk factor discussion is not relevant in this paper, because the paper does not address the issue of risk factors for breast cancer. 

"Study design and method

This section needs to include relevant statistical methods used for data analysis. It is recommended that use median and IQR for nonparametric data (age diagnosis) and use kruskal wallis test to measure the difference among age groups of diagnosed cases in case of scale measurements and use chi-square where percentages are compared. Further, Join-point regression can be used for trend analysis."

Authors: We have used the joint-point regression analysis. All other analyses were mainly descriptive. Classical tests such as Chi-square, Kruskal-Wallis oneway ANOVA are for hypothesis testing. Here, we do not test any scientific hypothesis, and those tests are not necessary. 

"Line 115-116 "we computed the point incidence of breast cancer for each 5-year period… 2010, and 2011 - 2015 inclusive". Briefly, mention how that incidence rate was calculated?

Line 118-119: "we calculated the age-standardized incidence rate for each of the 4 periods…" How age-standardized incidence rate was calculated?"

Authors: The calculation of age-standardized rate was based on the direct method of standardization, which is straightforward for basic epidemiologic studies. We however have described the calculation in the Methods section. 

"Line 119-121: "We also employed a segmented Poisson regression model to estimate the change in the incidence of breast cancer over time. All statistical analyses were conducted using the R Statistical Environment….".

It is recommended that use join-point regression technique and report your results in form of estimated annual percentage change (EAPC) with 95% UI. Please mention R version and package name that used for analysis." 

Authors: There are many methods to identify inflexion point in time series analysis, and joint point regression is one of them. We have now used the joint-point logistic regression analysis. 

"Line 125: findings (13,498 women, or 95%) and same in abstract, are inconsistent with results calculated in Table 1. (Table 1 has 13,948 women, 98%). Additionally report p-value of test difference and chi-square value."

Authors: Thank you for spotting this error. The actual number was 13,948 or 98% of total. There was no hypothesis in this study, and we don't think P-value is relevant here. 

"Line 133-138, "The average at diagnosis was 52 years (SD 11.6) and 56.3 years (13.4) for women and men, respectively. In women, there was a slight but statistically significant increase ……. (Table 2)"."

Authors: Thank you. We have now reported the trend. 

"Firstly, data is nonparametric so authors should report median with IQR for age diagnosis, rather than mean and SD. Author reported median age in abstract but results section included mean age? Please be consistent. Authors reported that "in women statistically significant increase in the average age of diagnosis of breast cancer during 1996 and 2015….". This difference look non-significant; please report p-value with test statistic value. Also, include these results in Table. Table 2 can be revised using joint-point regression estimate (EAPC with 95%UI for each duration)." 

Authors: We are not sure of the comment of "data is non-parametric'. In this dataset, the mean and median were highly comparable, and this is expected for a continuous variable such as age. However, we agree with the reviewer that the median is a better statistic. We have now reported the median age at diagnosis, and P-value based on the linear regression model. 

"Line:143-149: "Segmented regression indicated that there were two trends in the incidence of breast cancer in women: the first period occurred between ….(Figure 1). Further analysis showed that there was a statistically significant increase in the age-specific incidence of breast cancer over the period of 1996 and 2015, and the increase ….(Figure 2). In women, the increase in the age-specific incidence rate was observed among those aged….".

Join-point regression is widely used trend analysis technique and segmented regression is a part of it. Therefore, it is suggested that use main name of the technique for convenience of the readers."

Authors: We realize that joint-point regression is commonly used in trend analysis. However, this is a purely descriptive study, and we consider that such an analysis is not our main focus. A simple plot of incidence against year is good enough to see a change point. Nevertheless, we have now provided a result of joint-point regression analysis. We emphasize that our aim is to provide actual data on the burden of breast cancer; we are not interested in statistical exercises. 

"Figure 1 and 2 , are not readable. A better presentation is needed here. Author should draw the trends across ages, years and cohort by year and age group (e.g. within age Group, within year and within cohort). Through these 3 figures Table 2 results can be well representative."

Authors: We agree with the reviewer. We have removed both figures from the manuscript. 

"Line 160: Table 3. Briefly explain how the standardization was performed (ASR)?" 

Authors: This has been described in the Methods section. 

"Conclusion

Line 53-54: "These very first data from Vietnam suggest that although the incidence of breast cancer in Vietnam remains relatively low, it has increased over time, and that the increase was mainly attributable to those age groups of 50 and 70"."

"Finally, authors concluded that "Our data also confirm that Vietnamese women tend to have breast cancer at younger ages compared to Caucasian women". Younger ages? Conclusion is not consistent with the findings. (Further, most of the GBD studies reported that women breast cancer is more prevalent in older ages worldwide). Make any changes to the abstract that align with those made in the text."

Authors: We consider that our conclusion is consistent with the data. However, we have removed the sentence "and that the increase was mainly attributable to those age groups of 50 and 70". The median age at diagnosis of breast cancer in this study was 50, whereas in the United States and Australia, this figure is 56-62. 

"Minor comments

Line 34: Revise sentence structure (In line with the related literature, for example, see following literature)

1. (Nguyen TP, Luu HN, Nguyen MV, Tran MT, Tuong TT, Tran CT, Boffetta P. Attributable Causes of Cancer in Vietnam. JCO global oncology. 2020 Feb;6:195-204.

2. Nguyen SM, Deppen S, Nguyen GH, Pham DX, Bui TD, Tran TV. Projecting cancer incidence for 2025 in the 2 largest populated cities in Vietnam. Cancer Control. 2019 Jul 22;26(1):1073274819865274.

3. Pham T, Bui L, Kim G, Hoang D, Tran T, Hoang M. Cancers in Vietnam-burden and control efforts: a narrative scoping review. Cancer Control. 2019 Jul 17;26(1):1073274819863802.)"

Authors: We have mentioned the papers in the Introduction. 

"Line 51: age-standardized incidence rate, replace it with ASR as its already mentioned in the abstract."

Authors: Thank you. 

"Line 55: those age groups of 50 and 70 years…" 

"Line 60: 2019 showed that approx…….." 

"Line 61: improper sentence structure.. What is meant by was also the most..?"

"Line 69: is lower than in Caucasian populations…….. remove "in"."

"Line 71 & 72. Mention reference 5 once after the completion of related information."

"Line 73: women tends to have breast cancer in ……." 

"Line 125: write as 1st Jan. 1996……" 

"Line 127-128: Rewrite these lines" 

"Line 133: word age is missing, modify as … The average age at diagnosis was 52 years …." 

"Line 145: later instead of latter"

"158: repetitive words in one sentence"

"Line 189: revise the sentence"

"198: unable to understand the statement"

"Line 202: Asian women tends to …."

"Line 224: cancer incidence rates in Vietnam's urban population remain ….." 

Authors: Thank you so much for your helpful remarks. We have modified the sentences as indicated. We greatly appreciate the reviewer's meticulous comments and helpful suggestions.

---

## [Decision Letter · Decision Letter 1]

20 Jan 2021

PONE-D-20-16460R1

Trends in breast cancer incidence in Ho Chi Minh City 1996 - 2015: a registry-based study

PLOS ONE

Dear Dr. Ho-Pham,

Thank you for submitting your manuscript to PLOS ONE. After careful consideration, we feel that it has merit but does not fully meet PLOS ONE’s publication criteria as it currently stands. Therefore, we invite you to submit a revised version of the manuscript that addresses the points raised during the review process.

There are some more issues related to this manuscript according to the first reviewer that need to be addressed. There are also some writing corrections suggested by the second reviewer in the attached file.

We look forward to receiving your revised manuscript.

Kind regards,

Mohammad R. Akbari

Academic Editor

PLOS ONE

Reviewers' comments:

Reviewer's Responses to Questions

**Comments to the Author**

1. If the authors have adequately addressed your comments raised in a previous round of review and you feel that this manuscript is now acceptable for publication, you may indicate that here to bypass the “Comments to the Author” section, enter your conflict of interest statement in the “Confidential to Editor” section, and submit your "Accept" recommendation.

Reviewer #2: All comments have been addressed

Reviewer #3: All comments have been addressed

2. Is the manuscript technically sound, and do the data support the conclusions?

Reviewer #2: Yes

Reviewer #3: Yes

3. Has the statistical analysis been performed appropriately and rigorously? 

Reviewer #2: Yes

Reviewer #3: Yes

4. Have the authors made all data underlying the findings in their manuscript fully available?

Reviewer #2: Yes

Reviewer #3: Yes

5. Is the manuscript presented in an intelligible fashion and written in standard English?

Reviewer #2: Yes

Reviewer #3: Yes

6. Review Comments to the Author

Reviewer #2: The article remain several spelling and gramma error. Some technical error mention in commend should be addressed (attached track-change version).

Reviewer #3: The authors have improved the presentation of their work and answered most of my queries satisfactorily. However, I remain very concern about the following points. Therefore, before it can be published, the following points need to clarify in the manuscript.

1. Where authors use the word significant or statistical significant with the results reported, give the p-value or 95% uncertainty interval with that findings, e.g. line 50, 145.

2. For better presentation of results in table 2 it is suggest to authors, insert one column in Table 2 with p-value heading and report p value there for women and men separately, that will indicate the significant difference among periods by ages.

3. To make article results more interesting for readers, I would like to suggest to authors, also draw estimated annual incidence rate by cancer type, e.g. compare Malignant and other cancer type trend (like figure 1), If authors have sufficient data for type of cancer by period.

4. Line 48, Authors reported, “The ASR of breast cancer during 2011-2015 period was 21.5 cases per 100,000 women…”. Why the ASR reported in line 48 and in Table 3 are inconsistent, e.g. women ASR during 2011-2015. Brief clarification needed on how these ASR was calculated? Also need to clarify or fix this confusion.

7. PLOS authors have the option to publish the peer review history of their article (what does this mean?). If published, this will include your full peer review and any attached files.

Reviewer #2: **Yes: **Pham Quang Thai

Reviewer #3: No

---

## [Author Response · Author response to Decision Letter 1]

22 Jan 2021

"Reviewer #2: The article remain several spelling and gramma error. Some technical error mention in commend should be addressed (attached track-change version)." 

Authors: Thank you very much for your help and the track changes. We have gone through the changes and corrected them. We however want to retain some technical terms (eg 'inflexion point', 'ljr', "individual person's data"). 

"Reviewer #3: The authors have improved the presentation of their work and answered most of my queries satisfactorily. However, I remain very concern about the following points. Therefore, before it can be published, the following points need to clarify in the manuscript.

1. Where authors use the word significant or statistical significant with the results reported, give the p-value or 95% uncertainty interval with that findings, e.g. line 50, 145."

Authors: As we stated previously, this study aimed at describing the trend of breast cancer incidence over the 1996-2015 period; it was not designed to test any hypothesis that requires P-value. 

"2. For better presentation of results in table 2 it is suggest to authors, insert one column in Table 2 with p-value heading and report p value there for women and men separately, that will indicate the significant difference among periods by ages."

Authors: This table is purely descriptive. The table is not meant to test any hypothesis. We consider that the actual ASR data are much more important and much more informative than any P-value which is largely sample size dependent. 

"3. To make article results more interesting for readers, I would like to suggest to authors, also draw estimated annual incidence rate by cancer type, e.g. compare Malignant and other cancer type trend (like figure 1), If authors have sufficient data for type of cancer by period."

Authors: We appreciate your suggestion. However, more than 99% of the cases were malignant; only 0.3% were classified as in-situ. 

"4. Line 48, Authors reported, "The ASR of breast cancer during 2011-2015 period was 21.5 cases per 100,000 women…". Why the ASR reported in line 48 and in Table 3 are inconsistent, e.g. women ASR during 2011-2015. Brief clarification needed on how these ASR was calculated? Also need to clarify or fix this confusion."

Authors: The number quoted in the Abstract is the ASR per year, whereas the number quoted in table 3 refers to the 5-year ASR (2011-2015 period). For consistency, we have used the 5-year rate in the Abstract.

---

## [Decision Letter · Decision Letter 2]

27 Jan 2021

Trends in breast cancer incidence in Ho Chi Minh City 1996 - 2015: a registry-based study

PONE-D-20-16460R2

Dear Dr. Ho-Pham,

We’re pleased to inform you that your manuscript has been judged scientifically suitable for publication and will be formally accepted for publication once it meets all outstanding technical requirements.

Kind regards,

Mohammad R. Akbari

Academic Editor

PLOS ONE

Additional Editor Comments (optional):

Reviewers' comments:

Reviewer's Responses to Questions

**Comments to the Author**

1. If the authors have adequately addressed your comments raised in a previous round of review and you feel that this manuscript is now acceptable for publication, you may indicate that here to bypass the “Comments to the Author” section, enter your conflict of interest statement in the “Confidential to Editor” section, and submit your "Accept" recommendation.

Reviewer #3: All comments have been addressed

2. Is the manuscript technically sound, and do the data support the conclusions?

Reviewer #3: Yes

3. Has the statistical analysis been performed appropriately and rigorously? 

Reviewer #3: Yes

4. Have the authors made all data underlying the findings in their manuscript fully available?

Reviewer #3: Yes

5. Is the manuscript presented in an intelligible fashion and written in standard English?

Reviewer #3: Yes

6. Review Comments to the Author

Reviewer #3: (No Response)

7. PLOS authors have the option to publish the peer review history of their article (what does this mean?). If published, this will include your full peer review and any attached files.

Reviewer #3: **Yes: **Dr. Sumaira Mubarik

---

## [Editor Report · Acceptance letter]

1 Feb 2021

PONE-D-20-16460R2 

Trends in breast cancer incidence in Ho Chi Minh City 1996 - 2015: a registry-based study  

Dear Dr. Ho-Pham:

I'm pleased to inform you that your manuscript has been deemed suitable for publication in PLOS ONE. Congratulations! Your manuscript is now with our production department. 

Kind regards, 

on behalf of

Dr. Mohammad R. Akbari 

Academic Editor

PLOS ONE